# A Phenomenological Inquiry of the Shift to Virtual Care Delivery: Insights from Front-Line Primary Care Providers

**DOI:** 10.3390/healthcare12080861

**Published:** 2024-04-20

**Authors:** Gayle Halas, Alanna Baldwin, Lisa LaBine, Kerri MacKay, Alexander Singer, Alan Katz

**Affiliations:** 1Rady Faculty of Health Sciences, University of Manitoba, Winnipeg, MB R3E 0W3, Canada; 2Department of Family Medicine, Rady Faculty of Health Sciences, University of Manitoba, Winnipeg, MB R3E 0W3, Canada; 3Community Partner, Winnipeg, MB R3E 0W3, Canada; 4Department of Community Health Sciences, Rady Faculty of Health Sciences, University of Manitoba, Winnipeg, MB R3E 0W3, Canada

**Keywords:** primary health care, COVID-19, virtual care, patient-oriented, healthcare provider experience, clinical services, Telehealth, telemedicine

## Abstract

The rapid deployment of virtual primary care visits served as a first-line response to COVID-19 and can now be examined for insights, particularly as virtual care is playing an ongoing role in patient care and consultations. Input from primary care providers directly responsible for virtual care delivery is needed to inform policies and strategies for quality care and interactions. The overarching goal of this research study was to examine the use of virtual care as a mechanism for primary healthcare delivery. A phenomenological approach investigated the shift in primary care service delivery as experienced by primary care providers and initiated during the COVID-19 pandemic. Focus groups were conducted with primary care providers (n = 21) recruited through email, advertisements, and professional organizations, exploring how virtual care was delivered, the benefits and challenges, workflow considerations, and recommendations for future use. Integrating virtual care was performed with a great deal of autonomy as well as responsibility, and overwhelmingly depended on the telephone. Technology, communication, and workflow flexibility are three key operational aspects of virtual care and its delivery. Providers highlighted cross-cutting themes related to the dynamics of virtual care including balancing risk for quality care, physician work/life balance, efficiency, and patient benefits. Primary care providers felt that virtual care options allowed increased flexibility to attend to the needs of patients and manage their practice workload, and a few scenarios were shared for when virtual care might be best suited. However, they also recognized the need to balance in-person and virtual visits, which may require guidelines that support navigating various levels of care. Overall, virtual care was considered a good addition to the whole ‘care package’ but continued development and refinement is an expectation for optimizing and sustaining future use.

## 1. Introduction

There are many deep and lasting impacts of the COVID-19 pandemic, notably on healthcare delivery throughout the world. Physical distancing and reduced in-person clinical care prompted the rapid, widespread adoption and use of virtual care, defined broadly as “any interaction between patients and/or members of their circle of care, occurring remotely, using any forms of communication or information technologies, with the aim of facilitating or maximizing the quality and effectiveness of patient care” [1,2,3]. Prior to March 2020, the utilization of technology for patients to communicate with their providers in Canada was low [4,5]. Mohammed et al. [6] reported that an average of 6.5% of medical visits were conducted virtually prior to the pandemic. With the onset of the pandemic, an average of 66.4% of visits were virtual and almost 44% of practitioners anticipated continuing some kind of virtual health service delivery. For primary care specifically, studies have been predominant in Ontario, with reports of rapid shifts to virtual care and an increase from 7% to almost 86% of physicians providing virtual visits in the months prior to and following the pandemic onset. In Manitoba, our team found that a rapid shift occurred in the first three months of the pandemic onset—with 33% of visits being conducted virtually (primarily by telephone) and almost 97% of providers having conducted at least one virtual visit [7]. This mirrors findings from a 2021 survey of virtual care use in Canada but with variations expected in response to the different waves of COVID-19 prevalence [8]. The rapid shift to virtual care, particularly in primary care, led to necessary adaptations to usual practices and clinical workflow [9,10]. In primary care, many providers had limited knowledge or experience with the technologies, policies, and jurisdictional issues that virtual care presented [11,12,13].

In Canadian primary care settings, the most common application of virtual care has been by telephone as it is easy to use, relatively inexpensive, and relies on technology that is commonly available to providers and patients [8,14,15]. More recent literature exploring the modalities used to deliver virtual care in Canada and abroad indicates video technology, secure messaging, and telephone as the mechanisms of communication emerging from the COVID-19 pandemic experience [5,15,16,17]. Most research to date has focused on the mode of virtual care, and participants in the current study provided some descriptions of virtual care use (Figure 1); however, much less is known about the perspectives of providers as predominant ‘users’ of virtual care, particularly as it becomes more integrated into primary healthcare interactions [7,8,18,19,20]. Questions remain about the impact of virtual care on workflow, ideal applications, subsequent recommendations, and the potential for sustained virtual care options and approaches.

The rapid deployment of virtual primary care visits served as a first-line response to COVID-19 and can now be examined for insights, particularly as virtual care is playing an ongoing role in patient care and consultations [8,10,12,14,20,21,22,23,24,25,26]. Input from those directly responsible for the activities and processes in the delivery of healthcare is instrumental for co-developing policies and strategies to maintain quality primary care in virtual interactions. The overarching goal of this research study was to examine the use of virtual care as a mechanism for primary healthcare delivery during the COVID-19 pandemic. We used an exploratory mixed-methods approach for a comprehensive examination of all users and their effects on care [27]. Here, we report the in-depth exploration of the shift in primary care service delivery as experienced by primary care providers, particularly how implementing virtual care affects practice workflow, the changes or adaptations needed to support implementation, and the challenges and strategies for optimizing and sustaining virtual care in clinical practice.

## 2. Materials and Methods

### 2.1. Design

A phenomenological approach was used in this qualitative study to retrospectively explore primary care providers’ experiences and perspectives of implementing virtual care during the COVID-19 pandemic. Ethics approval was received from the University of Manitoba Health Research Ethics Board (File No. HS24197).

### 2.2. Participants and Recruitment

A purposive sample of primary care providers from clinical practice sites across the province of Manitoba, Canada were invited to participate in our study. Recruitment was facilitated through email, advertisements, and notification of the study through the Manitoba College of Family Physicians and Doctors Manitoba and in academic e-newsletters sent by the Department of Family Medicine at the University of Manitoba. There were no exclusion criteria as we were seeking all levels of virtual care use in rural and urban practice locations. Individuals who expressed interest in participating provided consent and were reminded to not disclose anything within the context of the discussion and assigned to an online focus group. Focus groups were capped at a maximum of five participants per group for optimal online interaction [28].

### 2.3. Data Collection

Focus groups were initiated approximately 13 months following the introduction of virtual care tariffs locally and moderated by GH and AB via Zoom videoconferencing. Participants completed a brief demographic and practice profile prior to engaging in a 60-min focus group discussion. The research team developed a focus group interview guide, which was informed by a previous phase of the study [27] where key constructs were identified through a literature review. The final questions in the guide were decided in consultation with the research team and a Community Advisory Committee [27,29]. The team considered the need to have easy-to-understand, single-construct, neutral questions that would initiate and encourage discussion. The discussion explored how virtual care was delivered, the benefits and challenges, workflow considerations, and recommendations for the future of virtual care (focus group guide provided in Section A.1). Participants were provided with a small honorarium for their participation. Each focus group session was audio-recorded and transcribed verbatim. A debriefing session was held after each focus group for researchers to reflect on the discussion, including where further data elicitation and probing questions were needed or where questions needed more clarity.

### 2.4. Data Analysis

Transcripts from each focus group were imported into the NVivo 12.0 software program. Two members of the research team (AB, KM) initiated the analysis once KM received training on coding. They analyzed transcripts independently and met frequently to discuss progress, resolve discrepancies, and achieve consensus throughout the analysis process. Open coding was used to assign a category to each unit of meaning, resulting in one code list. Thematic statements or sentence clusters relevant to the objectives and supported by focus group data emerged and were reviewed and further refined by two additional research team members (GH, LL) [30,31]. Overarching patterns from phrases and sentence clusters were organized and codes were collapsed, resulting in key themes derived through consensus and triangulated through comparison with the current literature [31].

## 3. Results

### 3.1. Demographic and Practice Characteristics

A total of 20 family physicians and 1 pediatrician (N = 21) from 16 clinical practice sites across the province of Manitoba, Canada participated in one of six focus group sessions. The mean age of participants was 46 years, with most (N = 19) identifying as female. The majority of participants described their practice site as urban, medium to large-sized, and working within a fee-for-service remuneration model. Most participants (n = 14) reported maintaining the same number of clinic days per week before and during COVID-19 (i.e., since March 2020). All reported conducting virtual visits daily with the number ranging from 1–35 visits in a typical day. More detailed demographic and practice characteristics are provided in Table 1.

### 3.2. Themes

Overall, six key themes were identified (Figure 2). Technology, Communication, and Workflow pertain to the operational aspects of virtual care, whereas Balancing Risk and Physician Work/Life Balance are two underlying themes that crosscut all three. Patient Benefit and Efficient Care was a theme repeatedly highlighted throughout discussions of virtual visits. A rich compilation of participants’ quotations to support each of the key themes is provided in the text, with additional illustrative quotes from each theme provided in (Appendix A and Section A.2).

#### 3.2.1. Technology and the Basic Tools Required

While virtual care has been described as using any form of technology for remote communication and interaction, the telephone was the most commonly reported mode of communication for virtual care. Participants described how patients seeking a consult would call the clinic and speak to reception staff, who generally provided an appointment time for the physician to call the individual back. Providers reported that the return phone calls were being received by individuals in their homes as well as in a range of other external locations (e.g., work settings). During the pandemic, there was one clinic that opened its doors to individuals who did not have access to a phone, and a phone call was arranged with the physician who was off-site. Several participants also commented on their ability to proactively provide follow-up advice or monitoring with brief phone calls where needed. While the use of video was initially less common, some patients were able to send photos from their phones (for example, for a skin rash). The providers briefly commented on their dependence on reception staff to coordinate the virtual calls, and the use of video was more complicated and, in some patient cases, required additional support and time from staff (Figure 1).

Primary care providers’ reliance on the telephone was driven by the ease of its access and use for patients and providers, given the rapid shift to virtual care and the limited time within which to plan and implement alternatives. However, even basic telephone infrastructure requirements did not easily accommodate the shift to virtual visits. Participants noted that incoming calls from patients were not getting through due to phone lines being used for virtual visits, and many clinics were not equipped with enough telephone lines for providers to conduct in-office virtual visits:

*When we built the clinic, we weren’t anticipating that we would be making phone calls from all the exam rooms. So we didn’t put phone, landlines in every room. So that was kind of a thing that we had to figure out*.(FG participant 501)

Despite the general ease and accessibility of the telephone to conduct virtual visits, providers commented on the barriers as well as the issues experienced using other forms of technology. Video conferencing and secure messaging are alternatives to the telephone; however, malfunctions including unstable internet connection(s) or issues with audio or video quality frequently occurred, and not only for rural or remotely located patients:


*…the bandwidth you need to get a good video…the picture is horrendous.*
(FG participant 102)


*The downside is that you need Wi-Fi and so in some areas, in some like more rural places, the connection’s not very good and it keeps freezing.*
(FG participant 102)

While virtual care was considered by most to be a good option during the pandemic when there was the need to physically distance, many participants questioned the overall equity and accessibility of virtual care for patients, repeatedly expressing socio-economic concerns. Further, costs incurred by clinics or providers to set up more advanced communication options was another concern, particularly with little direction or knowledge of the ideal applications for sustained use in primary healthcare:


*I think we really have to think about this from an equity lens… Our infrastructure is inadequate and inordinately expensive.*
(FG participant 402)

#### 3.2.2. The Essentials of Communication

Communication, in all its forms, plays a significant role in comprehensive patient healthcare. Participants acknowledged the art and science of communication between providers and patients, where information is not simply conveyed verbally:


*If somebody’s walking into your office and they look like they’re a good color and they look energetic, and you see that they’re not short of breath. You already know a lot of information. We don’t have that information [on the phone]. So, you have to ask them specifically about things. So, it’s just a different way of gathering information. It’s also more time consuming.*
(FG participant 303)

Providers discussed feeling a heightened sensitivity to the explicit and implicit messages communicated by patients and stressed the importance of a complete history with extensive questioning to compensate for visual and other cues that were not as apparent in a virtual care encounter.


*One of the biggest things I’ve noticed is that I’m getting better at asking more questions and the history has become more important, like it should be. We were told in medical school, if you don’t know, by taking the history pretty much what you’re going for, then you haven’t taken a good history. And I’m finding that’s really true that we do ask more questions now when we can’t actually physically see, and I think it’s good, a good skill.*
(FG participant 202)

Even as providers adapted their approach to deliver more effective virtual care, several participants emphasized the unique ongoing and often long-term nature of relationships that are formed with their patients, particularly in primary care. Several primary providers highlighted the importance of the personal and medical histories they share with individuals and their families, which often served to mitigate the limitations associated with communication through virtual means.


*I think though, because I have had a lot of my patients for a long time, I think that was a helpful factor because I knew when something wasn’t right and I knew most of my patients.*
(FG participant 602)


*Yeah, so that’s helpful if you know your patients. My patients have been following me for 15 years for the most part. And so I can tell if they’re not doing well, that’s definitely true. If it’s somebody that you’ve never met before though, you wouldn’t know if they were different. So that’s a challenge. It’s easier to do virtual care with people you have a relationship with.*
(FG participant 303)

Still, providers experienced ongoing communication challenges during the shift from standard clinical practices to virtual care. Participants articulated their thoughts on what they perceived to be the acceptable limitations of virtual care and the scenarios in which it was less suited:


*I do a lot of addictions work and that’s really hard to do over the phone. So being able to observe withdrawal symptoms and do urine drug screening and so that type of condition is really hard. And also people with severe mental health. So it’s really generally the communication’s more challenging, and especially if they have co-occurring disorders, ‘cause you don’t know whether they’re intoxicated or whether they’re … Is just, there’s so many cues that you need to see those people in person.*
(FG participant 601)


*…any form of language barrier can be really difficult… combined with hard at hearing.*
(FG participant 201)


*I think the other story that I’ll relate is in the situation where you’re concerned about the safety of the individual and the truth is whoever they are concerned about, they can’t really speak to it because that individual is in the home, right, wherever they are.*
(FG participant 402)

#### 3.2.3. Managing Flexibility and Workflow

The shift from in-person clinical care to virtual care during the pandemic prompted adaptations to usual practices and clinical workflow. Many providers discussed using virtual care to connect with and triage patients to determine next steps or whether further assessment was needed:


*If you needed to see a doctor, let’s talk about what the issue is on the phone first, and then we would triage-oh, that sounds like it needs a physical exam or, oh, you’re losing weight, yeah, come in, we need some objective measures.*
(FG Participant 201)

Relying on virtual interactions to curb viral transmission during the height of the pandemic created an opportunity to learn about integrating virtual care into clinic workflow. Several primary care providers commented on the advantages of conducting an initial screening appointment or gathering prerequisite information from patients in a timely manner.


*I’ll talk to them on the phone, get most of the history and then I’ll just say, okay, just book an appointment, come in and see me and we can do the exam.*
(FG Participant 103)

In using virtual care as a starting point in care delivery, providers found they could better manage the timing of their day and streamline workflow. The following example describes better accessibility to patients as both parties did not have to overcome the issue of travel time to the clinic:


*And so I’ll call people sometimes like half an hour before their visit is scheduled. And I’ll just say, hey, I’m having some time. I’m calling earlier, are you free to chat now? And most of the time they’re like, yeah, no problem. It’s great. Now I can get on with my day. And so, I can then shift a lot of my people earlier in the day and then that opens up spots later in the day that people pulling same day can book. And I can get through more people that way. And it just gives me, yeah, it gives me that flexibility to manage my day a little bit better because I can get ahead of schedule rather than being behind.*
(FG participant 103)

Focus group participants further elaborated on the way in which virtual care enabled more flexibility in terms of the timing of scheduled appointments and thereby helped improve accessibility as wait times lessened and more appointment slots became available.


*But I think we’re fitting, we’re able to fit in more people because as I said earlier, those home visits can be, they don’t have as much socializing or wandering in the hall or taking time to switch over patients and clean the room and those sorts of things. So yeah, I find it moves a little faster.*
(FG Participant 201)


*It’s very easy to get that appointment with me. I was only seeing about 22 to 24 people a day. Now I’m seeing 28 people a day. So it’s actually easier for them to get an appointment the same day or the next day.*
(FG Participant 401)

Overflow Clinics were a workflow adaptation discussed by the participants. During the pandemic when primary care providers were limiting their physical presence in clinics, there was a need for a contingency plan to see patients when physical assessments were required and to be responsive to patients’ immediate needs without having to request they rebook and wait for another appointment. As one provider described:


*We have something called overflow clinic. If we saw the patient in the virtual clinic, and we believe that the patient needs to have the specific exam done, as long as it is not urgent thing, we refer to overflow clinic. And a few physicians, including me…[see] other physicians’ patients in overflow, and then communicate to the referring physician for the follow-up management.*
(FG Participant 302)

The implementation of an Overflow Clinic model is one example of redesigning workflow to provide comprehensive primary care services in a timely and efficient manner. To overcome the limitations of virtual care (and where more in-depth assessment is needed), the concept of an Overflow Clinic was felt to be advantageous from a provider perspective and several participants indicated its utility in maintaining continuity of care, despite patients often seeing a colleague rather than their regular physician.

The rapid shift to virtual care during the pandemic meant equally rapid changes to primary care practices, many of which occurred without much thought to workflow integration. While some changes brought about by virtual care had clear advantages for both providers and patients, other newly introduced processes created inefficiencies or added administrative work that still required further planning or development to elicit a more seamless integration. Participants commented:


*There’s definitely been some extra administrative tasks when we’ve been kind of having to pivot and swing between in-person and virtual…*
(FG Participant 501)


*Yeah, So I talk to them on the phone and if I feel I’m not getting a good enough idea of what’s actually going on, then I’ll say, well, I think I really need to see that. And then I’ll tell them to call back to the clinic and talk to the receptionist and say, Dr. XX told me to have an in-person visit. And I’ll also send her through [the electronic medical record] …a high priority message. And so hopefully she sees the messages about the same time the person’s calling that tells her what I told the patient so that she knows that I know that they’re asking for an in-person visit. And so then she’ll book them in and often I’ll send her in that message an idea of how soon I think that person should be seen.*
(FG Participant 202)

#### 3.2.4. Balancing Risk

Balancing risk(s) when managing patients remotely was a reoccurring theme throughout all the focus groups. Providers commented on the difficulty of finding balance to ensure that a patient is not at serious or pressing risk of harm. In particular, when unable to observe evolving symptomology directly and only relying on a patient’s account, providers expressed feeling a sense of unease after a virtual visit.


*I’ve had some scary stories where people tell me something that sounds like absolutely horrific when they’re describing it and then when they actually come in, there’s like barely anything to see there, right? Like, so they’re trying to describe a rash or something is black and horrible looking and you’re thinking, oh my gosh, do they have something necrotic going on? Like what’s happening and then when you actually see it, it’s nothing. So I think sometimes people’s descriptions, … you can’t see anything, right. So you do bring them in and you’re like, oh yeah, that’s nothing. But you spend all this time worrying that you’re going to come and see a toe falling off and you see like a tiny little, like, they stubbed their toe and there’s a little red spot there.*
(FG Participant 501)


*I think maybe you’re stretching your comfort level a little tiny bit, but being reassured by, well… this is my patient…. I’ve given them good return to care directions or what to look for. They will come back to me if this is an issue and that’s the same kind of comfort emerge docs have. Like you see someone with a headache. Yeah, it could have been brain tumor you sent away, but another emerge doc will catch it when they come back because you’ve given them good return to care instructions.*
(FG Participant 201)

In response to the uncertainty often felt when providing virtual care, both the need for and the notion of developing parameters or guidelines for virtual care were addressed:


*I would say like certain things should always be in person, right? If somebody’s presenting complaint is again abdominal pain or whatever, that should be in person. I need to see you. I need to figure out what’s going on. But I think a lot of like, so maybe there should be some way of categorizing what things are okay to be virtual visits and what things should always be in person.*
(FG Participant 501)


*Just kind of thinking about quality of care, quality of visits, and like tiering those and then how to compensate them in the future when COVID, isn’t such a huge, like there’s no restrictions and that kind of thing. Like your fax prescription refills, that’s your bare minimum level of quality care where you look at their medical issues and say, okay, at a glance, I know this person from the past. Sure, it’s safe to refill their medications or no, it’s not... I can talk to you on the phone, kind of get an idea of how things are. But then I guess your highest level of care and your highest level of visit is going to be that in-person.*
(FG Participant 501)

#### 3.2.5. Physician Work/Life Balance

As virtual care became more commonplace in primary care because of pandemic restrictions, finding the right mix and optimal delivery of virtual care in the absence of formal guidelines afforded considerable flexibility for providers to see what best suited them. Focus group participants discussed their early attempts at trying out differing schedules to see what worked best.


*I used to do like one full day of virtual, now I’ve kind of done like, okay, I’m going to do morning virtual, then I’m going to do my patients in the in-person patients in the afternoon. So that’s where I’m at right now. And that seems to be working for me. But I like the ability to be flexible and change based on what your needs.*
(FG Participant 303)

Having more latitude to conduct virtual visits from home or trying different schedule configurations of in-person and virtual visits had the effect of improving work–life balance and well-being. Providers felt that virtual care options allowed new-found flexibility to attend to the needs of patients and manage their practice workload, even if they were away from the clinic or had to care for their own family or personal issues as required:


*…personally like my parents are getting into their elder years and there might be emergency times that I have to go home. And in the past that would’ve meant you abandon your clinic, your colleagues pick up the slack, you do what you got to do in your personal life, come back when you can. But I thought, wow, this could be great. It could open up the possibility for me not to leave gaps in my practice in the future.*
(FG Participant 201)


*If their kids have to stay home from school ‘cause they have a sniffly nose, they’re not completely out of commission. Like it’s not like the whole day is shot. They can still stay home and they could still do like 90% of the work that they were going to do anyways.*
(FG Participant 501)


*I’ve been able to be around my family more.*
(FG Participant 102)

*…this* [virtual care delivery] *is calm.*(FG Participant 203)

#### 3.2.6. Patient Benefit and Efficient Care 

While some providers questioned the accessibility of virtual care, citing health equity concerns, others felt that virtual care options enabled greater reach and connection with patients who had been hard-to-reach or mostly out of touch with primary care:


*I think specifically for me, if I think about my patients over the last year, I definitely talked to more young people who may be a bit more nervous about coming in and more like men in their 30 s to 50 s who maybe feel more comfortable phoning, who don’t want to maybe leave work and are okay to call on a lunch break. I think those are two demographics I’ve probably had more contact with, just like kind of thinking back, nothing scientific in that, but.*
(FG Participant 501)


*People were very grateful and happy that we would call them and be interested in looking after them.*
(FG Participant 301)

Focus group participants suggested that the efficiency and ease of virtual services in primary care benefitted patients too frail to attend clinic in person, individuals with mental health issues, those living in remote areas of the province, and even those who …wouldn’t necessarily come in because they couldn’t take time off work. (FG Participant 501) Several providers commented on the benefit of virtual care visits as an option for patients when it was not necessary to attend in-person visits.


*And we felt like then we were doing something for a vulnerable population because a large number of these people were just too frail to come in. And they had been too frail to come in actually for a long time, but before we didn’t have good other ways to deal with that.*
(FG Participant 301)

The telephone is advantageous for people who may have barriers to coming in person.


*So, for our population that would be… maybe severe mental health. So, if they have a severe anxiety disorder that precludes them from leaving their home or they’re really uncomfortable, coming in person for that reason or if they have physical mobility challenges, that would be another difficult thing for them, to attend in person.*
(FG Participant 601)


*And also for remote, I mean, I also do consult work and I was able to talk to people from all over Manitoba on video where normally they would be flying in or like waiting a long time. So, I mean, that’s amazing. That’s what… that was overdue, to be honest. That should have been already established.*
(FG Participant 303)


*[Patients] lose time from work, you have to find parking, you have to pay a meter or a taxi or whatever it is, and you go down there and you sit there and do a thing that you could have done on a video.*
(FG Participant 202)

Providers acknowledged the benefit of virtual care services in another way to also connect with their patients’ caregivers and other in-home supports who may not otherwise attend in-person visits. Participants noted that caregivers were more accessible during virtual visits and often added valuable history or context, particularly for patients with declining health, cognition, or the ability to communicate. One health provider commented:


*I’ve been able to support some families [dealing] with really bad dementia and it’s really nice to be able to phone because the only person I can communicate with is actually the caregiver, because one is now completely non-verbal and the other is so sporadically verbal, it’s just expletives and it really doesn’t add much to the conversation.*
(FG Participant 101)

Equally, virtual options helped enhance the reach of primary healthcare, affording greater opportunities for both providers and caregivers—formal and informal—to be present for patients and participate in discussions involving health needs and future care plans:


*Well, so… we’ve always used especially in the inpatient setting, a team-based approach. But I think that it’s made the workflow better for many of my colleagues to be able to join by say MS Teams and be seen on camera for the family. ‘Cause usually it’s the patient that’s in the room with us, so the family and I think also for the family’s sake, as part of that whole, as a key player in the decision-making, then having an elder who’s from their community join say by Teams or any other support person that they would value being there.*
(FG Participant 402)

In some instances, virtual care was felt to be a more efficient means of providing care to patients than some traditionally established processes:


*I actually stopped doing prescription refills by fax and I would just tell people, book a virtual visit and we’ll talk about your blood pressure. See what it’s like at home, instead of just renewing whatever was sent to me. I felt like it was a better patient care that way.*
(FG Participant 303)

#### 3.2.7. Scenarios for Virtual Care

The experience-based themes provide a general overview of key insights provided by the participants. In addition to the cross-cutting themes, the participants shared concrete examples of when virtual care may be best utilized. We have compiled several verbatim quotes suggesting several indications for virtual care (Table 2). Notably, these options are presented as general scenarios; participants suggested a blended approach, further recommending consideration of the unique needs of the patient.

## 4. Discussion

The rapid and largely unplanned shift from mostly in-person clinical health services to virtual care during the early part of the pandemic was a necessary and effective strategy to provide and maintain continuity in primary health care delivery. Primary care providers experienced a great deal of autonomy as well as responsibility for how they implemented virtual care and relied on their own resilience to ensure they upheld the quality of care for their patients.

Study participants mostly used the telephone due to the ease of its access and use, while also acknowledging the limitations of existing infrastructure. Similarly, other literature outlines the current limitations and underutilized modalities such as video technology and secure messaging, which are thought to be more resource-intensive, costly, and time-consuming to implement [6,15,16,20,32,33,34]. These challenges are compounded by other patient factors contributing to a widening “digital divide”, and socioeconomic and geographic barriers influence one’s access to a phone, cellular service, or network connectivity [35,36,37,38,39]. Practical solutions start with improving the overall technological infrastructure for providers and patients and developing tools that are seamless and compatible with EMR systems [20,35,40,41].

In addition to the infrastructure needed to operationalize virtual care, our findings highlight the profound impact of virtual care on communication in all its forms. In the absence of physical cues, such as facial expressions and body language, we found that providers felt challenged in new ways to gather information, communicate, and deliver quality care to their patients. First, virtual care appeared to be a suitable means for screening or initial assessment and as a form of triage to help determine the best options for care. Emphasis was placed on the need for PCPs to balance the risk to patients, especially given the challenges of managing patients’ concerns without a physical assessment [38,42,43]. Second, virtual care necessitated a different approach to communication, and study participants expressed their reliance on their medical training knowledge and extensive questioning and noted instances where virtual visits are less suitable, for example, when patients experience language or cognitive barriers. A similar qualitative study looking at patient and provider experiences of virtual care in Ontario and British Columbia identified language barriers and cultural differences as greatly impacting virtual care communication due to less ability to express concerns in a second language or cultural barriers that may affect knowledge of virtual care, including knowledge of one’s right and preferences [22]. Further, they and others have found that virtual visits changed the relationship between patients, caregivers, and providers, suggesting that patients may feel less connected to their providers during virtual care appointments [22,29,44]. In our study, providers acknowledged similar sentiments but emphasized the enduring therapeutic relationships unique to primary care that are formed with patients and their importance in mitigating some of the limitations of communication via virtual modalities.

The use of virtual care technologies necessitated adaptations to communication and clinic practices and prompted providers to redesign workflow to support a full spectrum of primary care services. The required adaptations and workflow changes have been the least described aspect of virtual care in the current literature [45,46]. Many of the changes to workflow occurred hastily, leaving providers to question accessibility, equitability, and quality of virtual care. Their ability to adapt and respond to the emerging issues of novel health service delivery added to their workload. Providers conveyed the importance of ensuring quality care but maintained some uncertainty around optimizing virtual care; many pointed to the need for guideline development to provide more structure for virtual care in terms of formal remuneration and integration into clinical practice and workflow. Aligning the provision of services with a method of delivery may need to be developed around a tiered approach to guide when a virtual visit could provide a suitable level of care and the varying time and services needed by each patient. There was consensus around the need to attend to ongoing development and optimal implementation strategies, including education for both clinic staff and patients and establishing virtual care within medical and residency training [45,46,47,48].

Despite some uncertainty with virtual visits, participants perceived greater reach to connect or reconnect with patients who had previously, for any number of reasons, been out of touch with primary care. Recent reviews by DeVera et al. [49], Mistry et al. [50], and Shaw J et al. [51] further describe key inequities in the access and delivery of virtual care and suggest strategies and practices to address these initially identified inequities in virtual care.

The importance of maintaining flexibility for providers to assess the suitability of in-person and/or virtual care, as well as input from patients as to their healthcare and virtual preferences, was a key finding. Participants were eager to garner input from patients about their preferences for virtual care and have them more involved in decision-making about the mechanisms used in the delivery of their care. A blended model that takes into consideration the patients’ and caregivers’ needs, overcomes limitations experienced by vulnerable populations, and offers an optimal workflow for providers while maintaining the safety and quality of primary healthcare services is ideal.

As “users” of virtual care, the providers in our study identified clear limitations across all themes; however, overall, the potential and benefit of primary care were highlighted. Following the precipitous adoption of virtual care and its now more sustained role within primary care, we gleaned a number of recommendations from the insights gained by providers directly responsible for the delivery of primary healthcare. Table 3 summarizes the key recommendations that emerged for integrating sustainable virtual care practice in primary healthcare.

A strength of this study is the application of qualitative methodologies to elicit rich descriptive data to provide further context and gain a better understanding of virtual care, where challenges can be mitigated, and the recommendations that can contribute to sustainable virtual care services well into the future. The perspectives gained from providers as one of the key “user’ groups of the healthcare system are needed, and together with findings from providers across Canada, constitute an important aspect for developing and optimizing policies and practices concerning virtual care.

This study had limitations and we acknowledge the value of feedback from other pivotal “user’ groups such as patients, interprofessional, and administrative staff, as well as residents in training. Their perspectives may be different from those included in this study. As part of the larger study, we conducted focus groups with patients, which are described separately. Our provider sample was comprised of mostly family physicians in addition to one pediatrician. While we were able to achieve theoretical saturation, the perspectives of virtual care are not fully representative of all primary care providers. Future research also needs to explore virtual care as a service delivery option among a range of healthcare providers and how it may influence team-based approaches to patient care. The providers that participated in this study had varying experiences with virtual care; however, no data were obtained from those who rejected or chose not to provide virtual care. Additionally, although the sample was mixed, it was predominantly female. The inclusion of more male-affiliated experiences may have resulted in different views and possibly different findings.

## 5. Conclusions

From primary care providers’ experiences and perspectives, virtual care was an effective mechanism for primary healthcare delivery in response to the COVID-19 pandemic. Six key themes were identified from the focus group participants regarding the operational aspects of virtual care (technology, communication, and workflow) along with the dynamics of balancing risk for providing quality care and the effect on physician work/life balance, efficiency, and benefits for the patient.

Integrating virtual care was performed with a great deal of autonomy as well as responsibility and overwhelmingly depended on the telephone. In the immediate term, this provided an accessible, reliable, and easy-to-adopt communication mechanism for maintaining continuity of care of patients. However, even telephone use required some basic technological infrastructure adaptations, suggesting further improvements would be needed for other options such as video teleconferencing or integration with electronic medical records.

Communication played a central role in patient care, relying on patients’ accounts of what was happening. Providers felt a heightened sensitivity to the explicit and implicit messages and the need for extensive questioning to compensate for the lack of visual and other cues. Therapeutic relationships and the awareness of personal and medical histories, as a hallmark of primary care, helped to mitigate the limitations of communication.

Primary care providers felt that virtual care options allowed increased flexibility to attend to the needs of patients and manage their practice workload, and a number of scenarios were shared for when virtual care might be best suited, including the initial assessment and triaging issues in advance of a clinic visit. However, they also recognized the need to have in-person options to respond to immediate needs.

Optimizing the process of virtual care may require guidelines that support navigating various levels of care. Patient and caregiver input is also a key element that will shepherd the use of virtual care that is responsive to a range of needs and specific circumstances. In essence, virtual care was considered a good addition to the whole ‘care package’, potentially offering convenience and accessibility, but continued development and refinement were an expectation for sustaining virtual care in the future.

## Figures and Tables

**Figure 1 healthcare-12-00861-f001:**
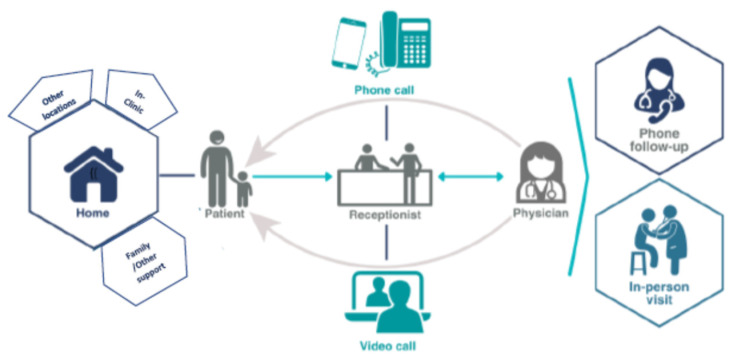
Virtual Care Processes.

**Figure 2 healthcare-12-00861-f002:**
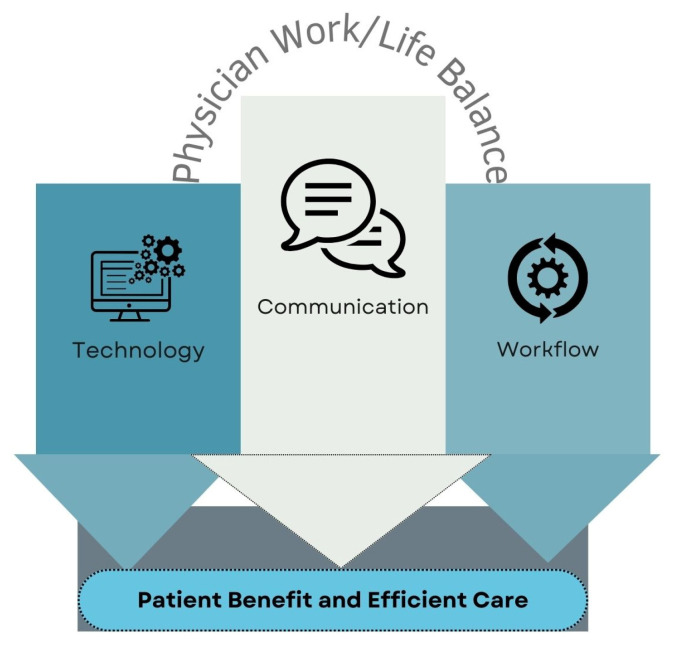
Key Themes.

**Table 1 healthcare-12-00861-t001:** Participant (N = 21) and practice characteristics.

Characteristics	N (%)
**Age**	
<35	3 (14)
35–49	11 (52)
>50	7 (33)
**Gender**	
Female	19 (90)
Male	2 (10)
**Geographic Area**	
Urban	15 (71)
Rural	5 (24)
Both	1 (5)
**Provider Type**	
Family Physician	20 (95)
Pediatrician	1 (5)
**Practice Size**	
<600	5 (24)
600–1000	8 (38)
>1000	7 (33)
Other	1 (5)
**Funding Model**	
Fee for Service	14 (66)
Alternative funded (salary, contract etc.)	6(29)
Stipend	1 (5)
**Number of PCPs in Your Clinic**	
<2	2 (10)
2–6	5 (23)
7–10	9 (43)
>10	5 (24)
**Number of Interprofessional Staff in Your Clinic**	
0–5	14 (67)
5–10	3 (14)
>10	4 (19)
**Number of Clinic Days Per Week (Pre-COVID Versus During COVID)**	
Stayed the Same	14 (67)
Increased	2 (10)
Decreased	5 (23)

**Table 2 healthcare-12-00861-t002:** Suggested scenarios for suitable virtual care.

Category	Verbatim Extract/Scenario
Educational health information	*I think that virtual care can be used a lot for like education, like for example, prenatal education.*
Continuing care/monitoring	*a blend of both in-person and virtual visits, I think is something which could be amazing. For simple refills or blood work requisitions or stuff like that, I think virtual is the way to go, but if it’s like more in depth conversation than the in-person would be the right way to do it.*
Continuing care/monitoring (chronic illness check-ins)	*I think a good blend of being able to do a few check ins a year for chronic health conditions, you don’t have to see them every visit.*
Educational health information/prevention consultations/mental and emotional health guidance	*how to manage anxiety, how to lose weight, how to … All these wellbeing things that you don’t have time to discuss*
Continuing care/follow up/monitoring	*follow-ups will probably just organically become virtual visits*

**Table 3 healthcare-12-00861-t003:** Key recommendations/key considerations (for VC future development).

Theme	Recommendation
Technology Infrastructure	Improve technological infrastructure for providers, including expanding the capabilities of the current tools available, develop seamless functions, and create compatibilities across EMR systems [35].
Communication Channels	Improve virtual care options that are accessible, equitable, and user-friendly for patients [36,37,38,39].
Workflow	Integrate virtual care options and services into the full suite of primary health care delivery (in both training and practice).
Balancing Risk	Develop a virtual care strategy or structured guide that is responsive to the breadth of patients and their health issues and considers a blended model of virtual and in-person care, tailored to one’s needs.
Physician Work/Life Balance	Promote flexibility to allow providers and patients to find the most suitable “blend” of virtual and in-person care.
Patient Benefit and Efficient Care	Consistent efforts to garner input from patients about their preferences for virtual care, with more patient involvement in decision-making about the mechanisms used in the delivery of their care.

## Data Availability

The data are not publicly available due to ethical and privacy restrictions. As previously stated, additional participant verbatim quotations within each theme are available as Appendix A.

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
