# Peer review of "A Phenomenological Inquiry of the Shift to Virtual Care Delivery: Insights from Front-Line Primary Care Providers"

_healthcare, 2024, doi:10.3390/healthcare12080861_

Round 1

Reviewer 1 Report

Comments and Suggestions for Authors

1. The first paragraph of the introduction should be summarized as “the importance of virtual nursing in primary health care”, followed by its drawbacks, which need to be re-summarized from line 54 to the following paragraph.

2. The introduction section on the current application of virtual care overlaps with the results and conclusions of this article, lacking innovation.

3. There is ambiguity in the position of the result section (N=21), and it is recommended to adjust it.

4. Is there a conflict between “everyone reporting virtual access every day” and “0-35 times”?

5. The reference to "language or cognitive impairment" in line 500 doesn't quite fit with the example given. Is it a cognitive impairment, a pathology, or a cultural difference? It is recommended to improve the expression method and further analyze the corresponding results section.

6. Please avoid duplicating content in the Discussion and Results sections. It should be an in-depth analysis and complement existing results.

Comments on the Quality of English Language

The language in the article is generally fluent, and it is recommended to modify the ambiguous parts.

Author Response

Dear Reviewer,

Comments and Suggestions for Authors

  1. The first paragraph of the introduction should be summarized as “the importance of virtual nursing* in primary health care”, followed by its drawbacks, which need to be re-summarized from line 54 to the following paragraph.

Response: This manuscript explored the shift in primary care service delivery as experienced by primary care providers; we have made it clear that the purposive sample included family physicians and a pediatrician. Thus, the requested change (i.e., “virtual nursing in primary health care”) is not accurate.

  1. The introduction section on the current application of virtual care overlaps with the results and conclusions of this article, lacking innovation.

Response: The introduction summarizes the ‘structures’ related to virtual care however much less is known about the fundamental processes, particularly adaptations and challenges experienced by the physician participants in providing health care services. This research provides a more in-depth analysis of these constructs with insights from front line primary care providers, generating a number of recommendations for future virtual care use and optimization.

  1. There is ambiguity in the position of the result section (N=21), and it is recommended to adjust it.

Response: Could we kindly request clarification on what the reviewer is suggesting regarding ambiguity in the position of the result section.

  1. Is there a conflict between “everyone reporting virtual access every day” and “0-35 times”?

Response: This was an error. It should read: “All reported conducting virtual visits daily with the number ranging from 1-35 visits in a typical day.”

  1. The reference to "language or cognitive impairment" in line 500 doesn't quite fit with the example given. Is it a cognitive impairment, a pathology, or a cultural difference? It is recommended to improve the expression method and further analyze the corresponding results section.

Response: The line reads as: “…instances where virtual visits are less suitable, for example where patients experience language or cognitive barriers.” The barriers have been more completely described in the results section and include language barriers (line 230), hardof hearing (line 230), frailty, mental health issues (line 404) and dementia (lines 434-442).

The line following the noted reference (line 500) is not an example but rather extends consideration of language and cultural barriers. We have added further information from the referenced study (Chan-Nguyen et al). The sentence now reads:

A similar qualitative study looking at patient and provider experiences of virtual care in Ontario and British Columbia identified language barriers and cultural differences as greatly impacting virtual care communication due to less ability to express concerns in a second language or cultural barriers that may affect knowledge of virtual care, including knowledge of one’s right and preferences.21

  1. Please avoid duplicating content in the Discussion and Results sections. It should be an in-depth analysis and complement existing results.

Response: We are unsure of where duplication exists except to briefly tie the specific results to other research/literature in the discussion.

Reviewer 2 Report

Comments and Suggestions for Authors

This is a very well-written and presented paper, providing clear findings and recommendations for virtual care. It is interesting how similar the findings are to eHealth scoping reviews completed prior to the pandemic. I have one minor suggestion and a second question.

First, I find Figure 1 to be distracting in it's current form. Perhaps this could be better visualized as a type of flow chart? The size and colours minimize the text in my opinion.

Second, I am surprised that there was no discussion on remuneration for virtual care. While not familiar with Manitoba, in other provinces the amount of remuneration was increased temporarily but then cut significantly. This has all but stopped most virtual care. Was this a theme that was discussed?

Author Response

Dear Reviewer,

Comments and Suggestions for Authors

This is a very well-written and presented paper, providing clear findings and recommendations for virtual care. It is interesting how similar the findings are to eHealth scoping reviews completed prior to the pandemic. I have one minor suggestion and a second question.

First, I find Figure 1 to be distracting in it's current form. Perhaps this could be better visualized as a type of flow chart? The size and colours minimize the text in my opinion.

Response: Thank you for this feedback! We have revised with the concept of a flow chart in mind, and with professional graphics that provide a more polished appearance.

Second, I am surprised that there was no discussion on remuneration for virtual care. While not familiar with Manitoba, in other provinces the amount of remuneration was increased temporarily but then cut significantly. This has all but stopped most virtual care. Was this a theme that was discussed?

Response: Interesting points however remuneration wasn’t a topic or theme that emerged during the focus groups. More specifically to your point, there was no similar increase and subsequent decrease in virtual care funding in Manitoba. Virtual care was unfunded prior to the pandemic in Manitoba (there are no corporate virtual care providers in Manitoba as a result). The funding did not change during the period when the experiences reported here occurred. It is therefore not surprising that the issue was not raised by the respondents.

Reviewer 3 Report

Comments and Suggestions for Authors

I would like to thank you authors for preparing the above manuscript 'Experiencing the Shift to Virtual Health Care Delivery: Insights 2 from Front Line Primary Care Providers' in which, the pros/cons of virtual health care are described in details through experience.

I found the methodology, presentation and discussion very sound and appropriate. However, one missing point is 'patient demographics' and 'patient level experiences and impact' of virtual health care among the participants in the study you conducted. It would be beneficial to know patient demographics and patient level experiences in identifying ways to better support patient sue and satisfaction. 

Additional comments:

Further elaboration on why phenomenological approach was chosen and how it aligns with the research objectives need to be clarified.

- Clarifying how participants were identified and selected, as well as any considerations regarding diversity or representation within the sample, would strengthen the methodology. Participants need to be from diverse backgrounds (across age, gender, rural/semi rural/urban, healthy/number of comorbidities etc.). Additional recruitment strategies (social media platforms or collaborating with regional healthcare associations or community groups) could help broaden the reach and diversity of participants and providers.

- Acknowledging potential biases or limitations associated with the research design and participant recruitment process would provide a more comprehensive understanding of the study's scope and findings.

In order to minimize response bias and ensure that participants provide honest and accurate responses during the focus group discussions, the authors could emphasize the confidentiality and anonymity of participant responses.

- The inclusion of 52 references may be considered somewhat excessive for this study. The conclusions are consistent with evidence presented but the volume of the sample in the study is very low and lacks diversity.

Author Response

Dear Reviewer,

Comments and Suggestions for Authors

I would like to thank you authors for preparing the above manuscript 'Experiencing the Shift to Virtual Health Care Delivery: Insights 2 from Front Line Primary Care Providers' in which, the pros/cons of virtual health care are described in details through experience.

I found the methodology, presentation and discussion very sound and appropriate. However, one missing point is 'patient demographics' and 'patient level experiences and impact' of virtual health care among the participants in the study you conducted. It would be beneficial to know patient demographics and patient level experiences in identifying ways to better support patient sue and satisfaction. 

Response: We appreciate the very positive response to this manuscript. We agree that the patient perspectives are important, which is why we designed an exploratory mixed methods approach for a comprehensive examination of all users and effects on care.*  A manuscript entitled “Patients’ and caregivers’ experiences of virtual care in a primary care setting during the COVID-19 pandemic: A patient-oriented research study” is currently being processed for publication ( DOI: 10.1177/20552076241232949).  In this current manuscript we report on the in-depth exploration of the shift in primary care service delivery as experienced by primary care providers. 

*Halas G, Singer A, Katz A, et al. Examining virtual visit use during a pandemic and perspectives of primary care providers, patients and caregivers: a mixed-methods research protocol.  BMJ Open 2022;12:e062807. doi: 10.1136/bmjopen-2022-062807

Additional comments:

Further elaboration on why phenomenological approach was chosen and how it aligns with the research objectives need to be clarified.

- Clarifying how participants were identified and selected, as well as any considerations regarding diversity or representation within the sample, would strengthen the methodology. Participants need to be from diverse backgrounds (across age, gender, rural/semi rural/urban, healthy/number of comorbidities etc.). Additional recruitment strategies (social media platforms or collaborating with regional healthcare associations or community groups) could help broaden the reach and diversity of participants and providers.

Response: It is always the hope of the research team to have an optimal recruitment outcome. As stated in lines 92 – 95, we used a variety of methods to reach a large (possibly the majority) of family physicians in our province.  We were very fortunate to have 21 individuals provide their time during a pandemic and health care resource crisis. 

This study benefitted from the data provided by providers who demonstrated a broad age range, with variable sizes in patient panels and clinic settings, and with ~25% of respondents being from rural areas (reported in Table1).  As stated in lines 562-570, future research would benefit from the input of providers who reject or choose not to provide virtual care as well as with more equitable representation from male providers.  Future research may also consider greater breadth in the research question and examine virtual team-based care as an interprofessional service delivery option.

- Acknowledging potential biases or limitations associated with the research design and participant recruitment process would provide a more comprehensive understanding of the study's scope and findings.

Response: Please note lines 562-571 outline limitations regarding generalizability. We achieved theoretical saturation with the data provided by this sample. This is exploratory work and as such, will benefit from further research in the key areas identified.

In order to minimize response bias and ensure that participants provide honest and accurate responses during the focus group discussions, the authors could emphasize the confidentiality and anonymity of participant responses.

Response: All participants provided consent (lines 97-98). The study consent form was reviewed prior to the focus group. We’ve added text (lines 98-99) which states: “…reminded not to disclose anything said within the context of the discussion.”  The consent form includes conventional wording stating that findings from this research study may be published or presented in public forums, however will not reveal names or other identifying personal information. 

- The inclusion of 52 references may be considered somewhat excessive for this study. The conclusions are consistent with evidence presented but the volume of the sample in the study is very low and lacks diversity.

Response: The sample size for a qualitative, exploratory study of providers’ experiences is not low, especially considering that four members of the research team participated in the analysis and reached theoretical saturation. We agree that the sample lacks some diversity -a point that is made in lines 562-571, which outline limitations regarding generalizability.

Reviewer 4 Report

Comments and Suggestions for Authors

Thank you for inviting me as a reviewer of this valuable manuscript. I recommend following suggestions for improving quality of manuscript.

Title

(Comment 1) I recommend authors to specify study design as a subtitle. The current subtitle is written like a review article.

Intruduction Section

(Comment 2) I recommend authors to describe your primary care before the COVID-19 pandemic and your primary care after it.

(Comment 3) I recommend authors to supplement the definition and explanation of Virtual Health Care. In particular, I recommend authors to add a figure (or graphic abstract) for this part. Also, the difference between ‘virtual health care’ and ‘virtual health care delivery’ needs to be explained.

Method Section

(Comment 4) I recommend authors to specify study design (lines 84-89). Is it a quailtitave study design? If so, I recommend authors to supplement with references relevant to the study design used.

(Comment 5) If the questionnaire has not been used in previous research, the questionnaire development process must be supplemented.

Result Section

(Comment 6) Figure 1.0 should be changed to Figure 1 (line 137). Figure 1 (Efficiency & Patient Benefit) is difficult to understand. (line 144). Keywords and explanations must also be added for the intersection part.

Discussion Section

(Comment 7) Through this focus group interview, a schematic that synthesizes the process, components, and suggestions for moving to ‘Virtual Care’ is needed.

(Comment 8) ‘Virtual Care’ is also related to telemedicine and home healthcare. Be sure to mention this in your review

Appendix Section

(Comment 9) I recommend authors to remove appendix in manuscript and provide it as a supplementary file.

Author Response

Dear Reviewer,

Comments and Suggestions for Authors

Thank you for inviting me as a reviewer of this valuable manuscript. I recommend following suggestions for improving quality of manuscript.

 Title

(Comment 1) I recommend authors to specify study design as a subtitle. The current subtitle is written like a review article.

Response: With the addition of the study design, we have revised the title to:

A Phenomenological Inquiry of the Shift to Virtual Care Delivery: Insights from Front Line Primary Care Providers

Introduction Section

(Comment 2) I recommend authors to describe your primary care before the COVID-19 pandemic and your primary care after it.

Response: This is an exploratory study of virtual care and not a pre-post intervention comparison.  Describing the providers’ experiences of virtual primary care in the face of the COVID-19 pandemic is precisely what this research was designed to do.

(Comment 3) I recommend authors to supplement the definition and explanation of Virtual Health Care. In particular, I recommend authors to add a figure (or graphic abstract) for this part. Also, the difference between ‘virtual health care’ and ‘virtual health care delivery’ needs to be explained.

Response: Thank you for your attention to the words used to describe our study purpose. For clarity, we have changed several lines in the title and manuscript and maintained use of the phrase “virtual care” rather than “virtual health care” to avoid any potential confusion. This is then more closely aligned with the definition provided in lines 52-54.

We appreciate your recommendation for a figure or graphic however this is beyond the scope of our inquiry and creativity.  The referenced definition of virtual care has been in use for some time and is a very straightforward description of the concept and breadth of modalities.

The added use of the term “delivery” may carry an intuitive understanding in some jurisdictions. However, according to the European Observatory on Health Systems and Policies, health care delivery is the more visible functions of the health system for public/patients. (See https://eurohealthobservatory.who.int/themes/health-system-functions/health-care-delivery). To provide clarification, the final intro paragraph now includes a sentence that reads:

Input from those directly responsible for the activities and processes in the delivery of health care is instrumental for co-developing policies and strategies to maintain quality primary care in virtual interactions.

 Method Section

(Comment 4) I recommend authors to specify study design (lines 84-89). Is it a qualitative study design? If so, I recommend authors to supplement with references relevant to the study design used.

Response: This is a qualitative study based on phenomenology.  Changes were made to line 86, and a reference added to line 87. 

(Comment 5) If the questionnaire has not been used in previous research, the questionnaire development process must be supplemented.

Response: A questionnaire was not used.  This is a qualitative study using a focus group interview guide which was developed in consultation with the research team and community advisory committee.  It was also informed by previous research. It is highly unusual to describe the process of developing an interview guide, as it is simply a guide and additional probing questions are used to prompt further exploration or to seek clarification or more detail around a response provided by a participants.  These are not completely developed a priori but emerge in response to what participants say during an interview albeit within the scope of the inquiry.

 Result Section

(Comment 6) Figure 1.0 should be changed to Figure 1 (line 137).

Response: Thank you and change has been made.

Figure 1 (Efficiency & Patient Benefit) is difficult to understand. (line 144). Keywords and explanations must also be added for the intersection part.

Response: Thank you for this feedback! We have revised with your wording suggestion in mind, and with professional graphics that provide a more polished appearance.

 Discussion Section

(Comment 7) Through this focus group interview, a schematic that synthesizes the process, components, and suggestions for moving to ‘Virtual Care’ is needed.

Response: We are not familiar with schematics for the focus group process. The narrative provides a rich description of what participants conveyed, particularly key messages regarding their experiences of providing virtual primary care. Could we kindly request clarification and a description of what would be included in a schematic and its fit with the scope of this work. 

(Comment 8) ‘Virtual Care’ is also related to telemedicine and home healthcare. Be sure to mention this in your review

Response: We have not conducted a review. As exploratory work, we need to be responsive to the data provided by the participants. Telemedicine may be related semantically, however the participants had a collective understanding of ‘virtual care’ which was used throughout. Home healthcare was not mentioned by the participants at all.

 Appendix Section

(Comment 9) I recommend authors to remove appendix in manuscript and provide it as a supplementary file.

Response: We are open to this modification and will accommodate editorial changes.

Round 2

Reviewer 4 Report

Comments and Suggestions for Authors

The authors did not addressed most of my comments. The answer is not only insufficient, but also insincere. Improvements in the revised paper are not revealed. For this reason, I decided to reject the paper.

Author Response

My response was in no way meant to be insincere, but was hoping for clarification and with the intent of staying true to the exploratory nature of the research.  A complete response is provided in the attachment.

Our team has been trying to provide a figure (as requested).  If the attached figure is suitable, we would be happy to refine further.
